# Identification of a Putative SARS-CoV-2 Main Protease Inhibitor through In Silico Screening of Self-Designed Molecular Library

**DOI:** 10.3390/ijms241411390

**Published:** 2023-07-13

**Authors:** Nanxin Liu, Zeyu Yang, Yuying Liu, Xintao Dang, Qingqing Zhang, Jin Wang, Xueying Liu, Jie Zhang, Xiaoyan Pan

**Affiliations:** 1School of Pharmacy, Health Science Center, Xi’an Jiaotong University, Xi’an 710061, China; xingerxueminglnx@stu.xjtu.edu.cn (N.L.); yzy10166@stu.xjtu.edu.cn (Z.Y.); liuyuying@stu.xjtu.edu.cn (Y.L.); dxt123@stu.xjtu.edu.cn (X.D.); zqq_fighting@stu.xjtu.edu.cn (Q.Z.); wangjin1221@stu.xjtu.edu.cn (J.W.); zhj8623@mail.xjtu.edu.cn (J.Z.); 2School of Pharmacy, The Fourth Military Medical University, Xi’an 710032, China; xyliu0427@163.com

**Keywords:** virtual screening, COVID-19, SARS-CoV-2 main protease, enzymatic assay

## Abstract

There have been outbreaks of SARS-CoV-2 around the world for over three years, and its variants continue to evolve. This has become a major global health threat. The main protease (M^pro^, also called 3CL^pro^) plays a key role in viral replication and proliferation, making it an attractive drug target. Here, we have identified a novel potential inhibitor of M^pro^, by applying the virtual screening of hundreds of nilotinib-structure-like compounds that we designed and synthesized. The screened compounds were assessed using SP docking, XP docking, MM-GBSA analysis, IFD docking, MD simulation, ADME/T prediction, and then an enzymatic assay in vitro. We finally identified the compound V291 as a potential SARS-CoV-2 M^pro^ inhibitor, with a high docking affinity and enzyme inhibitory activity. Moreover, the docking results indicate that His41 is a favorable amino acid for pi-pi interactions, while Glu166 can participate in salt-bridge formation with the protonated primary or secondary amines in the screened molecules. Thus, the compounds reported here are capable of engaging the key amino acids His41 and Glu166 in ligand-receptor interactions. A pharmacophore analysis further validates this assertion.

## 1. Introduction

Since the emergence of the coronavirus disease 2019 (COVID-19) in late 2019, it has become a significant public health concern, with confirmed cases reported across the globe, and continuing to rise. As of now, the World Health Organization (WHO) has recorded over 755 million confirmed cases, and a cumulative death toll of approximately 6.8 million [1]. Severe acute respiratory syndrome coronavirus 2 (SARS-CoV-2) has been identified as the primary pathogen responsible for COVID-19 [2,3,4], which is an RNA virus known to easily mutate into new variants with different characteristics [5,6,7]. Despite the widespread administration of vaccines, the latest variant, Omicron, has been able to evade the effects of the vaccine, and spread globally, resulting in fatalities [8,9]. Therefore, developing inhibitors against SARS-CoV-2 is of paramount importance to combatting the pandemic.

The proteolytic function of the main protease (M^pro^), also known as 3-Chymotrypsin-like protease (3CLpro), plays a critical role in the replication of SARS-CoV-2 during the initial stages of viral replication. The coronavirus genome is structured as a 5′-cap and a 3′-PolyA tail, and contains 6–12 open reading frameworks (ORFs). The first ORF (ORF1a/b) makes up roughly two-thirds of the genome’s length, and is responsible for producing two polyproteins, pp1a and pp1ab, via the a-1 frameshift between ORF1a and ORF1b. These polyproteins are then cleaved by the main proteases in host cells into 16 nonstructural proteins (NSPs), to generate functionally active viral replication complexes [10,11]. Therefore, developing inhibitors for M^pro^ can effectively hinder the replication and spread of the virus, and thus can play a crucial role in controlling epidemics [12,13,14].

The main protease (M^pro^) is a cysteine-catalyzed enzyme. Its Cys145 and His41 residues perform a nucleophilic addition reaction with the amide bonds of substrate peptides, promoting viral replication by cleaving them through hydrolysis (Figure 1A) [15,16,17]. Most developed inhibitors target this active site, to hinder the binding of the substrate to cysteine, and inhibit M^pro^ activity. For instance, the compound N3, which can covalently bind with Cys145, exhibits a high affinity and strong inhibitory effect with M^pro^, as screened by Zhenming Jin et al. [18]. Due to the presence of an α,β-unsaturated carbonyl group in the molecule (Figure 1B), it is capable of undergoing a Michael addition reaction with cysteine, resulting in the formation of a stable covalent bond. Adam G. Kreutzer et al. synthesized the cyclic peptide inhibitor UCI-1, which can stably bind at this location and inhibit the protein [19]. Similarly, Souvik Banerjee et al. screened FDA-approved drugs at this active site through high-throughput screening, and proposed that nilotinib, an anti-leukemia tumor drug, potentially has highly active M^pro^-protein-inhibitory effects [20].

Nilotinib is a second-generation inhibitor of the Bcr-Abl protein, a key kinase in chronic myeloid leukemia. In our previous research, our main focus was on the design and synthesis of inhibitors specifically targeting leukemia. Many compounds were derived from existing tyrosine kinase inhibitors, primarily nilotinib, incorporating molecular scaffolds, structural modifications, and optimization strategies [21,22]. Nilotinib primarily comprises an aromatic heterocyclic moiety containing a pyrimidine core, an aniline group substituted with trifluoromethyl and imidazole, and a linker that connects these two components (Figure 1B). Since the compounds we developed have similar structures to nilotinib, we speculate that they may also have the potential to inhibit M^pro^. We have built a molecular library of our compounds, and the structures of all the compounds are given in Appendix A. In this study, we employed virtual screening technology to evaluate the binding affinity of hundreds of our previously designed compounds with M^pro^, with nilotinib as the positive control, and Indole guanidine derivatives as the native control [23], in order to discovery a novel M^pro^ inhibitor (refer to Section 3.1) [24].

## 2. Results and Discussion

### 2.1. Virtual Screening

***Protein Activity Site Analysis.*** In this study, we obtained the M^pro^ structure from the PDB protein database (https://www.rcsb.org/structure/6LU7, accessed on 28 April 2022), and the native ligand for M^pro^ was the potential molecular inhibitor N3 [18]. Designed by Haitao Yang et al. in 2005 [25], N3 is a broad-spectrum inhibitor targeting the main protein of coronavirus. By binding to the active site of the SARS-CoV-2 main protease, and interacting with the Cys145 residue through a Michael addition reaction, N3 is capable of forming a stable covalent bond that exerts an inhibitory effect on the activity of the protein. The three-dimensional spatial structure and secondary structure of the main protein can be observed in 6LU7 (Figure 2A,B) [26,27,28,29], which consists of two domains, including 10 α-helices and 13 β-sheets. The ligand binds to the active site near β3, β4, and β16, with Cys145 and His41 as key residues.

***SP Docking.*** We conducted standard precision molecular docking (SP docking) on all 320 molecules, including the positive compound nilotinib, to assess their affinity to M^pro^ (Appendix A) within the active site. The molecular structures of the compounds were drawn using Maestro’s 2D Sketcher (version v128117, Schrödinger 2018-1, New York, NY, USA), and were subsequently prepared for docking through the use of LigPrep (LigPrep, Schrödinger, LLC, New York, NY, USA). Prior to docking, the protein underwent preparation and energy optimization, with the aid of Wizard in Glide. The receptor-grid-generation tool was employed to generate a grid box centered on the native ligand N3, with adjustments made to the grid size to optimize calculations. The minimized molecules were then docked to the grid box, utilizing standard precision docking. The results indicated that most drugs had a binding energy between −5.5 to −7.5 kcal/mol (Figure 2C), while 63 compounds exhibited a higher docking binding energy than nilotinib, with Δ*G* > −7.026 kcal/mol. (Table 1 and Appendix A).

***XP Docking and MM-GBSA Evaluation.*** As the extra-precision docking (XP docking) mode is more advanced, it helps to filter out false positives, and provides a better association between the docking score and good poses [30]. Therefore, XP docking was employed to re-dock the top 64 molecules, including nilotinib, with M^pro^, and to rank compounds according to their XP binding energy (Table 1). Based on the protein and re-docked ligands, molecular mechanics/generalized Born surface area (MM-GBSA) analysis was conducted, to evaluate the free ligand binding energy of 48 compounds with a higher XP binding energy than nilotinib (Δ*G* > 5.476 kcal/mol). The results of the analysis identified 32 molecules with higher free binding energies compared to nilotinib (dG > −48.96 kcal/mol), which were selected for further screening studies.

### 2.2. Further Screening through Induced-Fit Docking

The induced-fit theory was introduced by Koshland in 1995, and posits that enzymes do not have a complementary shape to the substrate, but form a complementary shape only after induction [31]. When it comes to ligand-receptor interaction, the conformation of the receptor, especially around the binding site, was also induced to be altered, better matching the shape and binding pattern of the ligand molecule. In this study, to obtain more realistic models of molecule-protein interactions, induced-fit docking (IFD) was applied for further virtual screening.

The 32 compounds were chosen to be re-docked into M^pro^ through the application of induced-fit docking (IFD), with nilotinib as the control, and their induced-fit-docking binding energy was calculated. Multiple docking results were produced during IFD docking, due to the various poses of the compounds, and we recorded the results with the highest docking energy for each compound in Table 1. The interactions of protein-ligand complexes were then displayed and analyzed using Maestro, and the potential interactions of all 32 compounds were counted (Figure 3, Figure 4, Figure 5, Figure 6, Figure 7 and Figure 8 and Appendix A). In comparison to the SP and XP docking, the IFD docking facilitated stronger binding and more protein interactions with the filtered molecules, further accentuating the differences in affinity between these compounds and M^pro^. As shown in Table 2 (or Appendix A), nilotinib interacts with M^pro^ mainly through hydrogen bonds, consistent with what has been reported in the literature. As for the screened compounds, they are able to engage more amino acids in protein-ligand interactions, and display new interactions, such as electrostatic interaction and pi-pi stacking hydrophobic interaction, compared to nilotinib.

Here, considering the diversity in structural features and binding interaction models, five molecules are selected as representatives to illustrate the molecule-protein interactions. Nilotinib was employed as a control, to further clarify the mechanism behind the compounds’ capacity to generate stronger interaction forces.

***Nilotinib and M^pro^ Interaction.*** Nilotinib can bind to M^pro^ well, in its active pocket (Figure 3). The complex consists of seven hydrogen bonds (Glu142, Cys145, His164, Glu166, Gln 189), including (1) two N atoms on the pyrimidine with Cys145, Asn142, and Gln189, respectively; (2) NH linked to the pyrimidine with Asn142; (3) CO in amide with Asn142 and Gln189; and (4) NH in amide with Glu166. The IFD-docking binding energy of stabilization is Δ*G* = −9.179 kcal/mol.

***Compound V253 and M^pro^ Interaction.*** The compound V253 has a pyridine biphenyl amide structure, with serine as a linker. It can form ten hydrogen bonds (Leu141, Gly143, Ser144, Cys145, His164, Glu166, Gln189, and Gln192), two pi-pi stacking interactions (His41), and a salt bridge (Glu166), which enhances the stability of the complex. In addition to similar hydrogen bonds to nilotinib, there are also other hydrogen bonds, including: (1) the NH of the amide in the serine backbone with His164 and Gln189; (2) the OH in the serine side chain with Leu141; (3) the N atom on the pyridine with Gln192; and (4) the NH of the amide linked to the pyridine with Glu166. Notably, V253 is capable of forming two pi-pi stacking interactions with the imidazole ring of His41. Furthermore, the acyl ethylenediamine connected to the pyridine contains a primary amine which has been protonated, allowing it to form salt-bridge interactions with Glu166. Overall, V253 could bind to M^pro^ through a hydrogen bond and pi-pi bond with His41 and Cys145, thereby impairing the activity of protease.

***Compound V247 and M^pro^ Interaction.*** The structure of the compound V247 closely resembles that of V253, with the only difference being the presence of a tBu protecting group on the serine sidechain of V247, thus making the docking pose and ligand-protein interactions similar to those of V253. The docking result of the compound V247 with M^pro^ shows six hydrogen bonds (Asn142, His164, Glu166, Gln189, and Gln192), and a salt bridge with Glu166, due to the protonated ammonia in the ligand, thus forming a stable complex. Exclusively, the CO of the amide in V247 generates one hydrogen bond with Asn142, which is different to that of V253. Interestingly, despite having fewer interactions, V247 demonstrates a similar binding energy to that of V253.

***Compound V133 and M^pro^ Interaction.*** The V133 shares a comparable structure with the compound V253, except for the linker type of amino acid present in its molecule, which is alanine instead of serine. Thus, the interaction type of V133 with M^pro^ is quite similar to that of V253. As shown in Figure 6, the compound V133 binds to the protein through six hydrogen bonds (Gly143, Cys145, Glu166, Arg188, and Gln189), and further promotes the stability of the complex through the salt bridge formed between the protonated ammonia (primary nitrogen in ethylenediamine) in the ligand, and Glu166. Uniquely, the CO of the amide attached to the pyridine demonstrates two hydrogen bonds, with Gly143 and Cys145, respectively.

***Compound V228 and M^pro^ Interaction***. The compound V228 has a backbone structure with tertiary leucine as a linker. The docking result shows that the compound V228 can bind to the active site of M^pro^, and their interaction includes seven hydrogen bonds (His41, Asn142, His164, Glu166, His172, and Gln189). Compared to V253, the incorporation of tert-butyl into the compound V228 confers a significant steric hindrance, leading to a distinct conformation in the docking results. The amide structure attached to the pyridine exhibits hydrogen-bonding interactions with His41, His164, and Gln189, while another amide structure linked to the halogenated benzene engages in hydrogen bonding with Glu166 and Asn142. Unlike the above compounds, V228 with a cyclopropanamide structure lacks a primary amine, so it cannot form a distinctive salt-bridging interaction with Glu166.

***Compound V291 and M^pro^ Interaction.*** Notably, the pyrrolidine of the proline in V291 provides the basis for restricting the deformation of the molecule during its binding to the protein, allowing its stable insertion into, and binding to, the protein active site. The indazole structure in V291 exhibits a stronger hydrophobic interaction, compared to the pyridine-linked pyrimidine structure in nilotinib, which facilitates ligand-protein interactions. This compound binds to M^pro^ through nine hydrogen bonds (Thr26, His41, Asn142, Gly143, Cys145, and Glu166), three pi-pi stacking interactions (His41 and His163), and one salt bridge (Glu166). Indazole has the unique ability to form two pi-pi stacking interactions with His41, which is not found in compounds with other structural types. The hydrogen bond between the ligand and Cys145 or His41 could prevent the key residues cysteine and histidine from participating in substrate hydrolysis catalysis. Three pi-pi stacking interactions improve the hydrophobic interaction, to enhance the binding of the complex.

Based on the results above, it was found that these screened compounds interact with the M^pro^ protein mainly through hydrogen bonding, pi-pi stacking, and salt bridges. Hydrogen bonding is the most common interaction in these molecule-protein complexes, and Gly143, Cys145, Glu166, and Gln189 are the key residues involved in hydrogen bonding interactions. Consistent with nilotinib, some compounds, such as V133, V253, and V291, can directly interact with cysteine 145 through hydrogen bonding, leading to the direct inhibition of the cysteine-mediated hydrolysis reaction. Moreover, His41 is commonly involved in pi-pi stacking with aromatic structures in docking molecules. In particular, the compound V291, with a larger indazole aromatic ring structure, exhibits stronger pi-pi stacking with His41, and hydrophobic effects, enabling His41 to participate more effectively in ligand-receptor interactions. Additionally, the secondary amine embedded in the proline of V291, and the primary amines of other compounds, facilitate the involvement of Glu166 in salt-bridge generation between the molecules and the M^pro^ protein. Notably, for Cys145 and His41, two key amino acids involved in enzymatic hydrolysis, nilotinib only forms hydrogen bonds with cysteine, while the screened compounds can simultaneously interact with another key amino acid, His41, in pi-pi stacking, which might enhance the enzymatic inhibition. Moreover, the reported compounds could generate salt bridges with Glu166, revealing a new bind mode with M^pro^. These findings could provide novel perspectives for the development of M^pro^ inhibitors.

### 2.3. Pharmacophore Analysis

On the basis of molecular docking, we counted and analyzed the molecules which could bind with M^pro^ at the active site. Next, these molecules were selected for superimposition through the application of their docking conformation (as shown in Figure 9A), and we further analyzed the pharmacophore of all the molecules. As shown in Figure 9B, there are seven common components identified from the superimposition: three aromatic rings (R10, R11, R12, orange), three acceptors (A2, A3, A4, pink), and one donor (D6, blue). In particular, R10 and R11, A3, and D6 are deeply embedded in the pocket of the active site, thereby providing steric hindrance to substrate binding to the protein. Besides, A2 and A4 are located near the key cysteine amino acid, to prevent the substrate from reacting with Cys145. Moreover, R12, which is frequently occupied by benzene rings, could exhibit hydrophobic interactions with the protein on the left side of the pocket. Meanwhile, R10s are predominantly captured by aromatic rings, allowing for effective pi-pi interactions with another crucial amino acid, His41. In addition, the amide structures are mostly observed at A3 and D6, with electron-absorbing groups promoting the polarization of the NH bond to effectively act as a hydrogen-bond donor, while the carbonyl oxygen acts as a hydrogen-bond acceptor, resulting in interactions with neighboring amino acids, such as Glu166. These findings are in agreement with the conclusions drawn from the ligand-receptor interactions demonstrated by the IFD-docking results.

### 2.4. Molecular Dynamics Analysis

The virtual screening and molecular docking have shown the binding ability of different compounds to the main protease, while molecular-dynamics simulations can provide insight into the dynamic stability of receptor-ligand complexes under physiological conditions. Thus, based on the binding affinity, ligand-receptor complex interaction, and structural diversity of the compounds (Appendix A), 20 compounds were selected (Appendix A) to undergo molecular-dynamics simulations, to investigate their binding stability (Figure 10). Throughout the molecular-dynamics simulation, one frame of trajectory was generated every 100 ps, resulting in 1000 frames after 100 ns of operation. Additionally, a thorough analysis of the resulting data was conducted, including the root-mean-square deviation (RMSD), the root-mean-square fluctuation (RMSF), and a protein-ligand contact analysis.

The conformation of the ligand-protein complex at 0 ns was used as the reference for the calculation of the RMSD values of all 1000 frames. During the simulation, six compounds (V131, V133, V172, V245, V247, and V253) showed poor dynamic stabilities (Figure 10), while other compounds exhibited stable binding with the M^pro^ protein at the active site. Fluctuations in the RMSD values are usually associated with changes in ligand-protein interaction. For example, the compound V247 displayed a large variation in RMSD; it initially formed a strong bond with the Glu166 of M^pro^, but this weakened or even disappeared after 27 ns (Appendix A). Appendix A shows the three-dimensional structure of the V247-M^pro^ complex, at 1 ns and 28 ns, respectively. It is evident that the N-(2-(dimethylamino)ethyl)nicotinamide of the compound V247 detached from the protein, and the ethylenediamine portion was entirely separated from the protein surface at 28 ns, rendering it unable to interact with residues, thereby leading to strong fluctuations.

As for the other 14 ligand-protein complexes with a good dynamic stability, they showed stable RMSD values over long periods of time. Four representative complexes were selected for further MD simulations over 100 ns, to verify the stability of the complexes (Appendix A). To delve deeper into the dynamic changes in protein-ligand contact during the simulation, an analysis of the root-mean-square fluctuation (RMSF) and protein-ligand contacts was conducted (Appendix A). The RMSF is useful for characterizing local changes along the protein chain, which helped us to identify the residues responsible for altering the fluctuations in the protein-ligand complex structure. For all complexes except the V222-M^pro^ complex and the V254-M^pro^ complex, the protein RMSF values exhibited a negligible variation, suggesting that the system was in a state of equilibrium throughout the simulation. The residues in contact with the ligand are annotated in green, and the status of their contact with the ligand was monitored every 0.2 ns, throughout the 20 ns dynamic simulation. Most compounds maintained contact with a number of particular residues during the dynamic simulation, except for the V75-M^pro^ complex, the V97-M^pro^ complex, and the V282-M^pro^ complex.

### 2.5. ADME/T Prediction

ADME/T is a very important standard evaluation in contemporary drug design and drug screening. Any compound that exhibits drug-likeness must have moderate ADME/T properties. QikProp was used here to predict the ADME/T properties of the 20 compounds mentioned above (Table 3). The QikProp analysis includes the following standard limits: the PSA (the Van der Waals surface area of polar nitrogen and oxygen atoms and carbonyl carbon atoms), QPlogS (the aqueous solubility, log S. S in mol dm^-3^ is the concentration of the solute in a saturated solution that is in equilibrium with the crystalline solid), QPlogPo/w (the octanol/water partition coefficient), donorHB, accptHB, CNS (the central nervous system activity), #metab (the number of likely metabolic reactions), human oral absorption, QPlogBB (the brain/blood partition coefficient), QPPMDCK (the apparent MDCK cell permeability in nm/s), QPPCaco (the apparent Caco-2 cell permeability in nm/s), and QPlogHERG (the IC_50_ value for the blockage of HEGR K+ channels). Based on the collective data, we inferred that the ADME/T properties of compounds were within the prescribed limits for a potential candidate drug compound.

### 2.6. Compound Enzymatic Activity Assay

Based on the MD stimulations and the ADME/T predicted results, nine compounds were selected for assessment for their inhibitory potency, and for the calculation of their IC_50_ values (Table 4). We used a fluorescence resonance energy transfer (FRET)-based assay to measure these 9 compounds’ inhibitory activity against the M^pro^ protein in vitro. As shown in Table 3, the compound V291 exhibited a potent inhibitory activity, with an IC_50_ value of 2.77 ± 0.56 μM, while other compounds did not exhibit significant inhibitory activity against M^pro^, within a concentration of 20 µM. This preliminary result provides biological evidence that the compound V291, which has a similar structure to nilotinib, could potentially exert an inhibitory effect on M^pro^.

## 3. Materials and Methods

### 3.1. Experimental Procedures

We have confirmed the structures of more than 300 compounds, and have compiled a database. The design concepts of the compounds, as well as the synthetic routes, have been reported in our previously published literature [21,22]. To identify compounds with a higher binding affinity than nilotinib, we used SP docking, resulting in the identification of 63 molecules. Following the SP docking results, a pharmacophore analysis was performed, to identify and examine the common features of the compounds’ docking conformations. Furthermore, we redocked these 63 compounds utilizing XP docking, and MM-GBSA analysis was further used to screen a total of 32 molecules with a good binding affinity. Induced-fit docking (IFD) was performed to identify the interactions between these 32 molecules and M^pro^. Based on the ligand-receptor interactions, and structure type (Figure 11 and Appendix A), 20 ligand-receptor complexes were subjected to a molecular-dynamics (MD) simulation to assess their stability, and the ADME/T properties of these 20 molecules (Appendix A) were evaluated. After careful evaluation, nine compounds with a good MD stability and reasonable ADME/T predictions were selected, and their IC_50_ against the M^pro^ protein was evaluated.

### 3.2. Ligand Preparation

All of the molecular structures of compounds were drawn in Maestro’s 2D Sketcher (Maestro v128117, Schrödinger 2018-1, New York, NY, USA). The molecular preparation and energy minimization were carried out under the OPLS3 force field, using LigPrep in Glide (Glide v91117, Schrödinger 2018-1, New York, NY, USA). With the help of Epik, the algorithm simulated the ionization state of molecules in the environment of pH = 7.0 ± 2.0 (Epik v56117, Schrödinger 2018-1, New York, NY, USA). The other parameters were maintained at the default, and each molecule generated, at most, 32 stereoisomers with retaining specified chiralities.

### 3.3. Protein Preparation and Grid Generation

The protein we used in this study was the SARS-CoV-2 main protease, and the X-ray co-crystal structure was provided by the PDB network database (https://www.rcsb.org/structure/6LU7, accessed on 28 April 2022). The protein was processed and optimized using Protein Preparation Wizard in Glide (Glid v91117, Schrödinger 2018-1, New York, NY, USA). Specifically, the protein was preprocessed to initially screen for problems and defects in the spatial structure, and then hydrogen atoms were added through H-bond assignment. After the removal of the water or any other molecular solvent from the structure, the protein was optimized under the OPLS3 force field. The other parameters were maintained at the default. We then obtained the minimized protein structure, to generate the grid file for ligand docking. The grid box was generated with the native ligand in 6LU7 as the center, within the receptor grid generation, and its size was adjusted for the best calculation range.

### 3.4. Molecular Docking

The minimized molecule was docked using ligand docking in Glide (Glid v91117, Schrödinger 2018-1, New York, NY, USA). All molecules bound to the active site of the protein through SP/XP docking and flexible docking. Post-docking minimization was used to optimize the ligand-protein complexes, limiting the number of optimized ligands to less than 6. “Sample nitrogen inversions” and “Sample ring conformations” were selected, with “Flexible” ligand sampling. A bias sampling of the torsions for all the predefined functional groups was performed, through adding the Epik state penalties to the docking score. Any constraints were ignored. The docking results were analyzed and displayed in Maestro. The docking affinity energy of the compound with the protein was calculated using the following equation:(1)$$\Delta G=-RT\;\ln K_{d}$$
where *R* is the Boltzmann gas constant (*R* = 1.987 cal/mol/K), *T* is the default temperature of simulated docking (*T* = 298 K), and *K_d_* is the binding affinity of the docking.

### 3.5. MM-GBSA

To evaluate the free binding energy between the protein and the docked ligand, the MM-GBSA of the Prime module (Prime v64117, Schrödinger 2018-1, LLC, New York, NY, USA) was employed [32]. The MM-GBSA dG (Molecular Mechanics/Generalized Born Surface Area dG) of the optimized receptor-ligand complex was calculated to determine the ligand-binding affinities. During the calculation of the free binding energy, the VSGB solvation model and the OPLS3e force field were applied. The binding energy was calculated based on the following equation:(2)$$\Delta G=E_{Complex\left(Minimized\right)}-\left[E_{Ligand\left(Minimized\right)}+E_{Receptor\left(Minimized\right)}\right]$$

The MM-GBSA calculations entailed maintaining the rigidity of all protein atoms, whilst relaxing the compound’s atoms. Furthermore, the ranking of the protein-compound complexes was carried out using binding-free-energy calculations.

### 3.6. Induced-Fit Docking (IFD)

The proteins and ligands were prepared in Maestro, as mentioned in the method above, followed by the induced-fit docking (Induced Fit Docking, Schrödinger2018-1, LLC, New York, NY, USA). The prepared ligands file was imported, and the standard protocol was selected, with the OPLS3 as the force field due to its wider parameter range and significant improvements. The native ligand was selected as centroid, to generate a box of automatically generated size. We ignored constraints and, in the Ligand options, we ticked “Sample ring conformations”, with 2.5 kcal/mol as the energy window. In Prime Refinement, we refined residues within 5.0 Å of ligand poses; with Optimize, “Side chains” was ticked; and in Glide Redocking, we choose “SP docking” for the precision. We redocked the structures within 30.0 kcal/mol of the best structure, and within the top 20 structures overal1. Other parameters were maintained at the default. The docking results were displayed in Maestro (Maestro v128117, Schrödinger 2018-1, New York, NY, USA), and their ligand-receptor interaction was analyzed, including hydrogen bonds (within 3.5 Å), halogen bonds (within 3.5 Å), salt bridges (within 5.0 Å), pi-pi stacking (within 5.5 Å), and pi-action (within 6.6 Å).

### 3.7. Pharmacophore Analysis

Based on the docking results of all of the compounds, a pharmacophore analysis was performed in Maestro. The protein-ligand complexes were imported, and used to create a pharmacophore model within the Develop Pharmacophore Model module (Phase, Schrödinger 2018-1, LLC, New York, NY, USA). “R (Aromatic Ring)”, “A (H-bond acceptor)”, and “D (H-bond donor)” were picked to be highlighted as features, and then a receptor-based excluded-volume shell was created, using the default parameters.

### 3.8. ADME/T Prediction

The drug-likeness of all the compounds was evaluated using QikProp (QikProp v68117, Schrödinger 2018-1, LLC, New York, NY, USA), to determine their absorption, distribution, metabolism, excretion, and toxicological (ADME/T) properties. The analysis results of the 20 selected compounds are shown in Table 3.

### 3.9. Molecular Dynamics Analysis

We carried out molecular-dynamics (MD) simulation studies, using the Dynamics module (Desmond v53011, Schrödinger 2018-1, LLC, New York, NY, USA) of Schrodinger. At the beginning, the complex was processed using the Protein Prepare Module, according to the default parameters. The bonding information was corrected, hydrogen atoms were added, and water molecules were removed, in order to obtain the minimized protein structure. Next, using the System Builder plate, the processed protein was imported, to be solvated with the water model of TIP3P and the force field of OPLS3. In order to ensure that the complex was completely wrapped in the simulated solvent environment, we set some parameters of boundary conditions, including the box shape of “Orthorhombic”, the box-size calculation method of “Buffer”, and “Minimize Volume”. The Na+ was added to neutralize the negative charge of the protein. Then, we used molecular dynamics to simulate the dynamic simulation of the complex, with default parameters. Finally, we used the function of the Simulation Interactions Diagram to analysis the output of the molecular dynamics. The root-mean-square deviation (RMSD) was used to measure the average change in displacement of a selection of atoms, for a particular frame concerning a reference frame. The RMSD value at any time can be calculated using the following equation:(3)$$RMSD_{X}=\sqrt{\frac{1}{N}\overset{N}{\underset{i=1}{\sum }}{\left(r_{i}^{\prime }\left(t_{x}\right)-r_{i}\left(t_{ref}\right)\right)}^{2}}$$
where *N* is the number of atoms in the atom selection; *t_ref_* is the reference time (typically, the first frame is used as the reference, and it is regarded as time *t* = 0); and *r′* is the position of the selected atoms in frame *x* after superimposing on the reference frame, where frame *x* is recorded at time *t_x_*. The procedure is repeated for every frame in the simulation trajectory.

The RMSF value at any time can be calculated using the following equation:(4)$$RMSF_{i}=\sqrt{\frac{1}{T}\overset{T}{\underset{t=1}{\sum }}<{\left(r_{i}^{\prime }\left(t\right)-r_{i}\left(t_{ref}\right)\right)}^{2}}$$
where *T* is the trajectory time over which the RMSF is calculated; *t_ref_* is the reference time; *r_i_* is the position of residue *i*; *r′* is the position of the atoms in residue *i* after superposition on the reference; and the angle brackets indicate that the average of the square distance has been taken across the selection of atoms in the residue.

### 3.10. In Vitro Enzymatic and Inhibition Assay

The half-maximal inhibitory concentration (IC_50_) assay was performed based on the fluorescence resonance energy transfer (FRET) effect [33,34,35,36,37]. The fluorescent peptide MCA-AVLQSGFR-Lys(Dnp)-Lys-NH_2_ was used here as the substrate. The quantities of enzyme and substrate used, as well as the reaction times, were optimized, to meet the requirement that the enzyme activity be in a linear phase. The M^pro^ proteins (0.2 μM) were added to 20 mM Tris-HCl (pH 7.3) with 150 mM NaCl, and added to a black 96-well plate, at 91 µL/well. The compounds were dissolved in DMSO, and gradient-diluted with PBS, and then added to a plate at 5 µL/well, and incubated with M^pro^ at 37 °C for 30 min. Then, 20 μM substrates were added into each well, with 4 µL/well. Positive controls (consisting of only the enzyme and substrate, without an inhibitor) and blank controls (containing only the substrate, without the enzyme) were also included in the experiments. Next, the samples were incubated in the dark at 37 °C for 10 min, and the OD intensities were then read at λ_ex_ = 320 nm and λ_em_ = 405 nm, using a microplate reader (Thermo Scientific, Waltham, MA, USA). Prism software was used to calculate the IC_50_ value of the compounds in the inhibition of main protease activity. Due to the instantaneous initiation of the reaction upon the substrate addition, it is essential to perform the operations as rapidly as possible, to avoid the complete catalysis of the substrate. The pre-experiment of this study has confirmed that the enzymatic catalysis time for all the substrates is sufficient to complete all operations, after the substrate addition (t > 15 min).

## 4. Conclusions

COVID-19 has become a global public health crisis that continues to threaten the lives of people worldwide. The main protease (M^pro^), which is critical in the proteolytic processing of polyproteins, and facilitating viral assembly, has gained much attention as a drug target. In this study, we aimed to find a potential inhibitor for the M^pro^ of SARS-CoV-2, based on the reported inhibitor molecule nilotinib. We designed and synthesized over 300 compounds with structures similar to nilotinib, and screened them using molecular-docking technology (SP, XP, and IFD) and MM-GBSA analysis, to choose the potential compounds. We investigated the binding stability between the compounds and proteins using molecular dynamics, and predicted the drug-likeness of the molecules using QikProp. After assessing the stability and ADME/T predictions, nine compounds were identified as having strong potential, and were chosen for subsequent bioactivity evaluations. Our results show that the compound V291 is the most promising inhibitor, with an IC_50_ value of 2.77 ± 0.56 μM. Additionally, in silico simulations revealed that His41 and Glu166 could engage in interactions with ligands, through pi-pi stacking and salt bridges, respectively. The pharmacophore analysis further confirmed these findings, which required the inhibitors possessing aromatic moiety and protonated nitrogen atoms to interact with His41 and Glu166. These findings provide valuable insights into the interactions between Bcr-Abl protease inhibitors and SARS-CoV-2 M^pro^, thus serving as a promising reference for the development of novel anti-SARS-CoV-2 agents. The compound V291 could be a potential candidate for further optimization as an M^pro^ inhibitor.

## Figures and Tables

**Figure 1 ijms-24-11390-f001:**
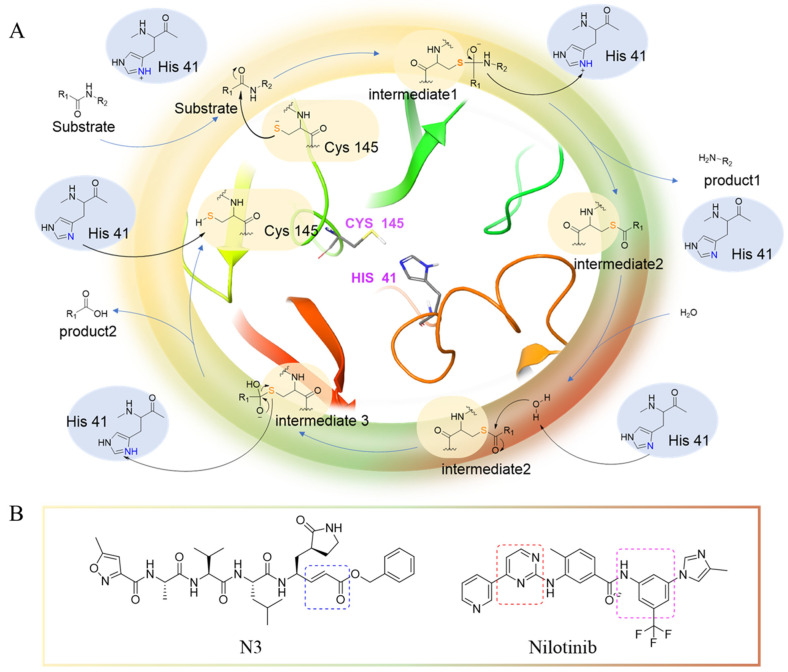
(**A**) The mechanism of cysteine kinase M^pro^ catalyzing the hydrolysis of substrate peptides. (**B**) The structures of N3 and nilotinib. The blue box corresponds to the structural fragment of the N3 molecule involved in covalent binding with cysteine. The red and purple boxes represent the aromatic heterocyclic moiety and aniline group, respectively, of the nilotinib.

**Figure 2 ijms-24-11390-f002:**
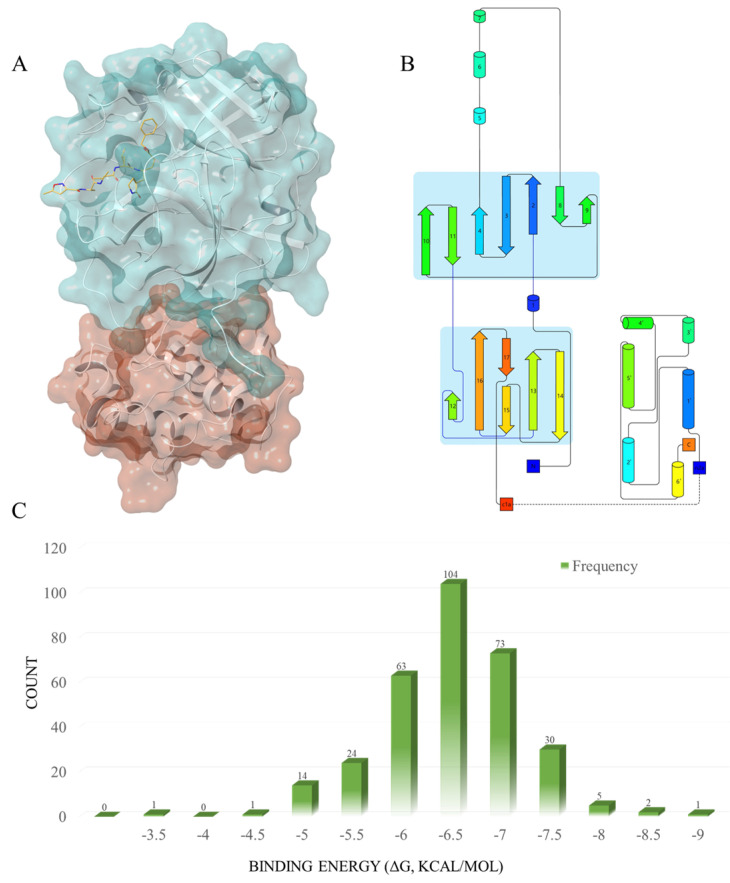
(**A**) The three-dimensional spatial structure of the main protease; Domain 1 (blue) and Domain 2 (red), with the native ligand N3 (yellow). (**B**) The secondary structure of the main protease. This figure was produced using the Pro-origami website (http://munk.cis.unimelb.edu.au/pro-origami/porun.shtml, accessed on 30 June 2016). The barrels represent α-helices; the arrows represent β-sheets. (**C**) The frequency distribution of the compound docking scores, based on the SP docking results.

**Figure 3 ijms-24-11390-f003:**
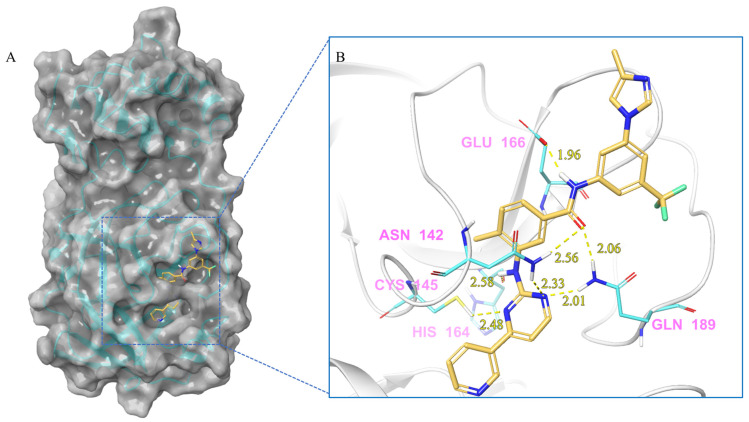
(**A**) The complex formed by the compound nilotinib and M^pro^. (**B**) The interaction between the compound nilotinib and the M^pro^ residues.

**Figure 4 ijms-24-11390-f004:**
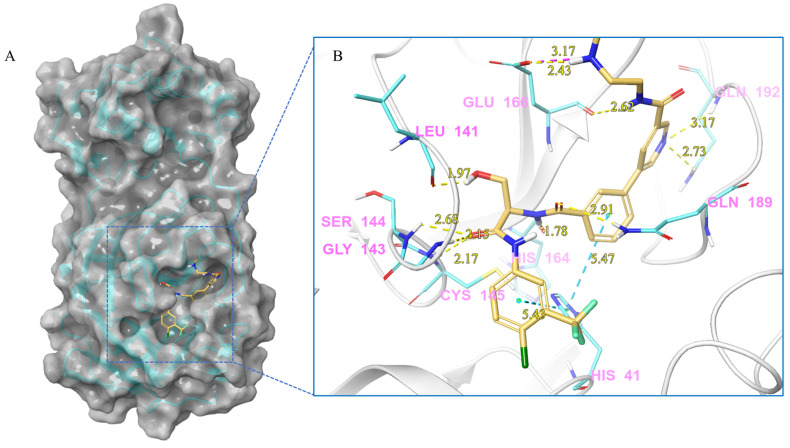
(**A**) The complex of the compound V253 binding with M^pro^. (**B**) The interaction between the compound V253 and the M^pro^ residues.

**Figure 5 ijms-24-11390-f005:**
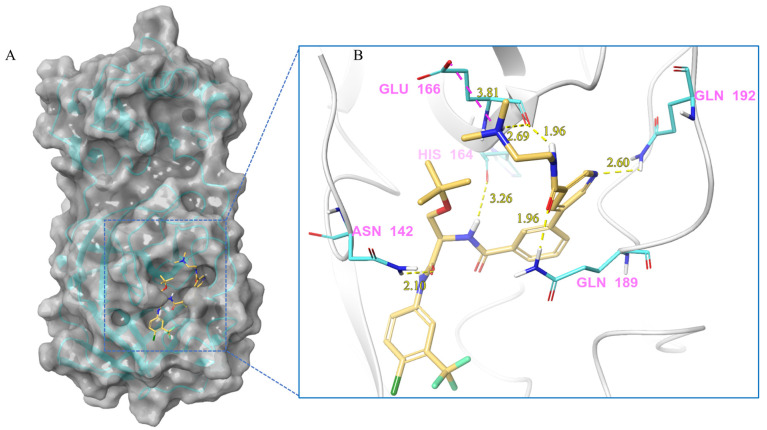
(**A**) The complex of the compound V247 binding with M^pro^. (**B**) The interaction between the compound V247 and the M^pro^ residues.

**Figure 6 ijms-24-11390-f006:**
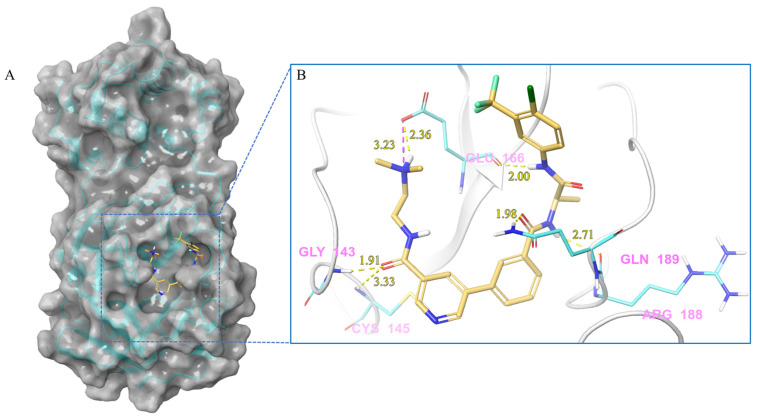
(**A**) The complex of the compound V133 binding with M^pro^. (**B**) The interaction between the compound V133 and the M^pro^ residues.

**Figure 7 ijms-24-11390-f007:**
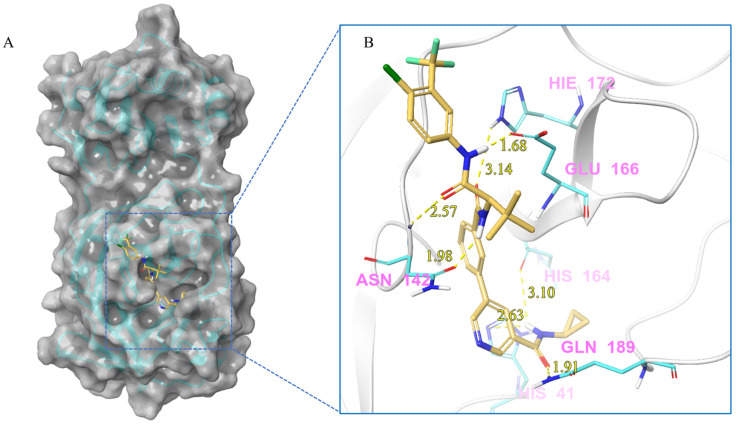
(**A**) The complex of the compound V228 binding with M^pro^. (**B**) The interaction between the compound V228 and the M^pro^ residues.

**Figure 8 ijms-24-11390-f008:**
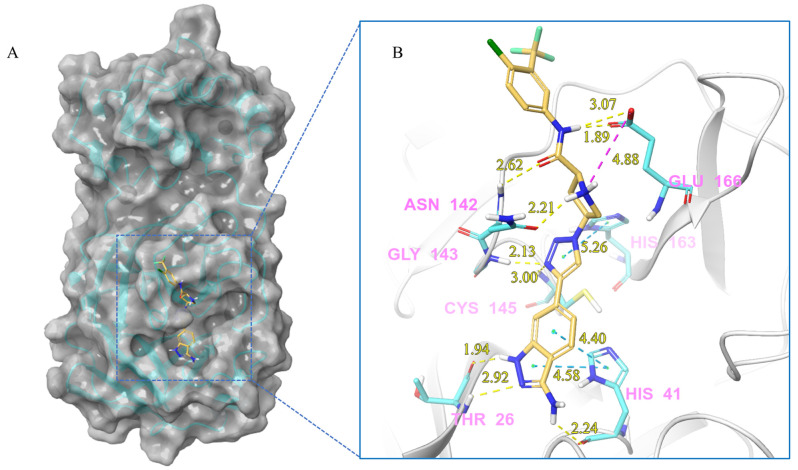
(**A**) The complex of the compound V291 binding with M^pro^. (**B**) The interaction between the compound V291 and the M^pro^ residues.

**Figure 9 ijms-24-11390-f009:**
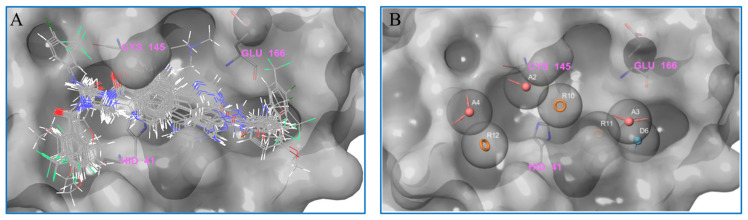
Analysis of the molecular pharmacophore. (**A**) The molecules binding at the active site show a certain regularity. (**B**) The common and important pharmacophore, obtained from the statistics of the molecular pharmacophore.

**Figure 10 ijms-24-11390-f010:**
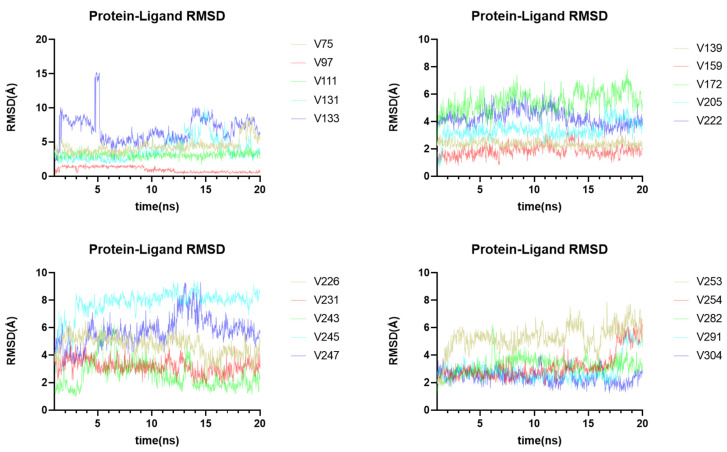
Molecular dynamics analysis of 20 ligand-receptor complexes. The RMSD value reflects the stability of a ligand-receptor complex.

**Figure 11 ijms-24-11390-f011:**
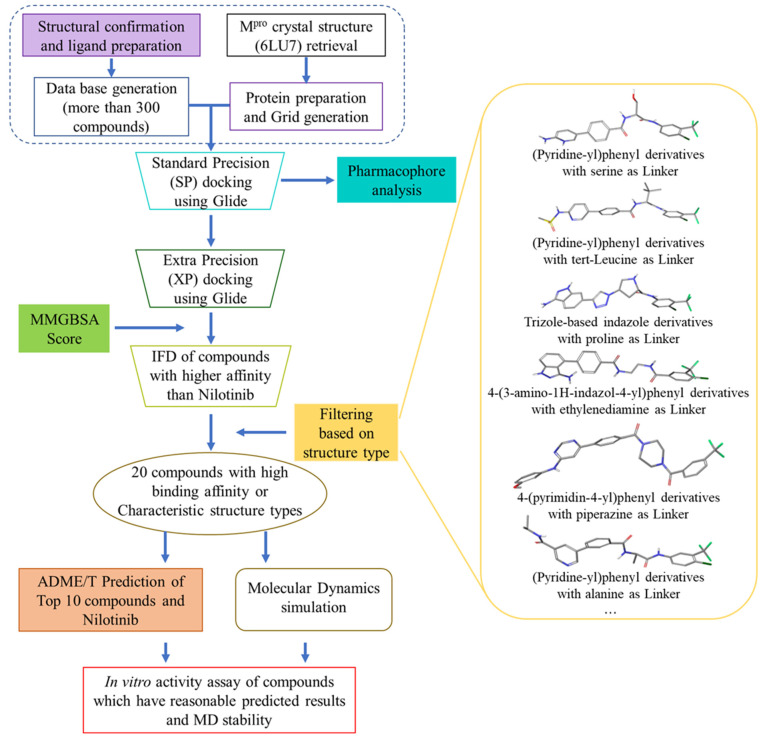
The flowchart of steps to identify the novel potential inhibitor.

**Table 1 ijms-24-11390-t001:** Compounds with a higher binding energy than nilotinib (ranked according to the XP docking results).

Rank	Compound	SP Docking Binding Energy (ΔG, kcal/mol)	XP Docking Binding Energy (ΔG, kcal/mol)	MM-GBSA dG Bind (kcal/mol)	IFD Binding Energy (kcal/mol)
1	V247	−8.538	−7.976	−63.43	−11.34
2	V253	−8.832	−7.97	−62.15	−12.787
3	V133	−8.379	−7.969	−58.29	−10.76
4	V109	−7.567	−7.108	−68.64	−12.342
5	V212	−7.688	−7.038	−47.32	-- ^a^
6	V131	−7.876	−6.759	−67.51	−9.919
7	V254	−7.286	−6.665	−57.63	−10.896
8	V248	−7.565	−6.645	−46.39	-- ^a^
9	V282	−7.481	−6.564	−49.38	−9.959
10	V139	−7.422	−6.49	−54.66	−9.584
11	V231	−7.372	−6.453	−57.18	−10.961
12	V128	−7.264	−6.422	−59.99	−9.557
13	V160	−7.241	−6.409	−58.19	−10.191
14	V163	−7.052	−6.337	−52.59	−9.817
15	V174	−7.369	−6.293	−56.94	−10.993
16	V243	−7.68	−6.267	−60.64	−11.459
17	V229	−7.509	−6.237	−57.48	−9.365
18	V204	−7.89	−6.229	−53.3	−9.652
19	V291	−8.071	−6.225	−59.08	−9.628
20	V228	−8.122	−6.12	−56.37	−10.792
21	V170	−7.085	−6.118	−50.4	−10.11
22	V215	−7.032	−6.08	−45.32	-- ^a^
23	V165	−7.051	−6.078	−48.72	-- ^a^
24	V75	−7.331	−6.072	−56.99	−10.714
25	V205	−7.501	−6.029	−47.89	-- ^a^
26	V120	−7.081	−6.025	−51.7	−9.171
27	V222	−7.305	−6.018	−48.47	-- ^a^
28	V245	−7.401	−6.008	−56.32	−10.098
29	V226	−7.141	−6	−61.64	−11.774
30	V159	−7.091	−5.995	−46.69	-- ^a^
31	V12	−7.192	−5.987	−48.94	-- ^a^
32	V225	−7.068	−5.983	−48.31	-- ^a^
33	V252	−7.416	−5.981	−57.86	−11.603
34	V219	−7.212	−5.972	−43.13	-- ^a^
35	V173	−7.138	−5.965	−47.29	-- ^a^
36	V241	−7.108	−5.963	−39.62	-- ^a^
37	V154	−7.26	−5.93	−49.83	−10.761
38	V304	−7.878	−5.911	−54.47	−9.512
39	V230	−7.314	−5.891	−43.13	-- ^a^
40	V172	−7.403	−5.85	−50.46	−11.391
41	V97	−7.138	−5.825	−60.03	−11.778
42	V111	−7.195	−5.823	−49.62	−10.706
43	V238	−7.134	−5.811	−55.55	−10.605
44	V155	−7.113	−5.77	−48.22	-- ^a^
45	V112	−7.29	−5.744	−44.36	-- ^a^
46	V60	−7.261	−5.637	−49.07	−10.191
47	V257	−7.37	−5.637	−48.64	-- ^a^
48	V103	−7.535	−5.55	−60.98	−9.795
49	Nilotinib	−7.026	−5.476	−48.96	−9.179
50	V74	−7.225	−5.262	-- ^a^	-- ^a^
51	V175	−7.538	−5.258	-- ^a^	-- ^a^
52	V283	−7.532	−5.159	-- ^a^	-- ^a^
53	V306	−7.335	−5.136	-- ^a^	-- ^a^
54	V293	−7.205	−5.042	-- ^a^	-- ^a^
55	V286	−7.432	−5.041	-- ^a^	-- ^a^
56	V168	−7.182	−5.023	-- ^a^	-- ^a^
57	V144	−7.195	−4.988	-- ^a^	-- ^a^
58	V122	−7.095	−4.935	-- ^a^	-- ^a^
59	V150	−7.145	−4.932	-- ^a^	-- ^a^
60	V240	−7.211	−4.738	-- ^a^	-- ^a^
61	V70	−7.299	−4.731	-- ^a^	-- ^a^
62	V86	−7.085	−4.152	-- ^a^	-- ^a^
63	V147	−7.364	−3.493	-- ^a^	-- ^a^
64	V303	−7.458	−3.164	-- ^a^	-- ^a^

^a^ Not measured due to low XP binding energy with M^pro^.

**Table 2 ijms-24-11390-t002:** The interactions between each molecule (five typical representatives) and M^pro^.

Compound	Molecular Interactions	Nature of Interactions	Distance (Å)
Nilotinib	Asn142:HD22-Lig:O1	Hydrogen Bond	2.56
	Asn142:HD21-Lig:N6	Hydrogen Bond	2.33
	Cys145:HG-Lig:N5	Hydrogen Bond	2.48
	Lig:H16-His164:O	Hydrogen Bond	2.58
	Lig:H9-Glu166:OE2	Hydrogen Bond	1.96
	Gln189:HE21-Lig:N6	Hydrogen Bond	2.01
	Gln189:HE22-Lig:O1	Hydrogen Bond	2.06
V253	Lig:N1-Glu166:OE2	Salt bridge	3.17
	Lig:H19-Leu141:O	Hydrogen Bond	1.97
	Gly143:H-Lig:O4	Hydrogen Bond	2.15
	Ser144:H-Lig:O4	Hydrogen Bond	2.68
	Cys145:H-Lig:O4	Hydrogen Bond	2.17
	Lig:H15-His164:O	Hydrogen Bond	1.78
	Lig:H28-Glu166:OE2	Hydrogen Bond	2.43
	Lig:H11-Glu166:O	Hydrogen Bond	2.62
	Gln189:HE21-Lig:O2	Hydrogen Bond	2.91
	Gln192:H-Lig:N3	Hydrogen Bond	3.17
	Gln192:HE21-Lig:N3	Hydrogen Bond	2.73
	His41-Lig	Hydrophobic (pi-pi Stacking)	5.47
	His41-Lig	Hydrophobic (pi-pi Stacking)	5.43
V247	Lig:N1-Glu166:OE2	Salt bridge	3.81
	Asn142:HD21-Lig:O4	Hydrogen Bond	2.10
	Lig:H15-His164:O	Hydrogen Bond	3.26
	Lig:H11-Glu166:O	Hydrogen Bond	1.96
	Lig:H36-Glu166:O	Hydrogen Bond	2.69
	Gln189:HE21-Lig:O1	Hydrogen Bond	1.96
	Gln192:HE21-Lig:N3	Hydrogen Bond	2.60
V133	Lig:N3-Glu166:OE2	Salt bridge	3.23
	Gly143:H-Lig:O2	Hydrogen Bond	1.91
	Cys145:H-Lig:O2	Hydrogen Bond	3.33
	Lig:H28-Glu166:OE2	Hydrogen Bond	2.36
	Lig:H24-Glu166:O	Hydrogen Bond	2.00
	Lig:H5-Arg188:O	Hydrogen Bond	2.71
	Gln189:HE21-Lig:O1	Hydrogen Bond	1.98
V228	Lig:H1-His41:NE2	Hydrogen Bond	2.63
	Asn142:H-Lig:O2	Hydrogen Bond	2.57
	Lig:H10-Asn142:OD1	Hydrogen Bond	1.98
	Lig:H1-His164:O	Hydrogen Bond	3.10
	Lig:H21-Glu166:OE1	Hydrogen Bond	1.68
	His172:HE2-Lig:O3	Hydrogen Bond	3.14
	Gln189:HE21-Lig:O1	Hydrogen Bond	1.91
	Lig:H1-His41:NE2	Hydrogen Bond	2.63
V291	Lig:N2-Glu166:OE2	Salt bridge	4.88
	Lig:H14-Thr26:O	Hydrogen Bond	1.94
	Thr26:H-Lig:N7	Hydrogen Bond	2.92
	Lig:H12-His41:O	Hydrogen Bond	2.24
	Lig:H19-Asn142:OD1	Hydrogen Bond	2.21
	Asn142:H-Lig:O1	Hydrogen Bond	2.62
	Gly143:H-Lig:N5	Hydrogen Bond	2.13
	Cys145:H-Lig:N5	Hydrogen Bond	3.00
	Lig:H2-Glu166:OE1	Hydrogen Bond	1.89
	Lig:H2-Glu166:OE2	Hydrogen Bond	3.07
	His41-Lig	Hydrophobic (pi-pi Stacking)	4.40
	His41-Lig	Hydrophobic (pi-pi Stacking)	4.58
	His163-Lig	Hydrophobic (pi-pi Stacking)	5.26

**Table 3 ijms-24-11390-t003:** The ADME/T prediction results of 20 compounds.

Compounds	PSA	QPlogS	QPlogPo/w	donorHB	accptHB	CNS	#metab	Human Oral Absorption	QPlogBB	QPPMDCK	QPPCaco	QPlogHERG
V75	109.835	−7.648	4.646	1	10.25	−2	2	1	−1.295	318.027	664.465	−7.395
V97	108.109	−7.351	4.096	1	10	−2	2	1	−1.35	469.306	338.538	−7.096
V111	122.61	−5.7	3.621	5	6.5	−2	1	3	−1.388	324.488	274.769	−6.644
V131	118.105	−9.052	5.627	2.25	8.25	−2	4	1	−1.183	1829.769	487.844	−7.478
V133	121.912	−7.656	4.886	2.25	10.25	−1	5	1	−0.944	458.546	123.765	−8.379
V139	131.795	−7.531	4.288	4	10	−1	4	1	−0.999	332.432	91.695	−8.504
V159	89.316	−8.698	6.02	1	8.5	−1	1	1	−0.568	3877.557	1007.174	−7.822
V172	89.548	−10.162	6.837	1	8.5	−1	3	1	−0.58	7550.982	849.164	−8.139
V205	111.144	−6.49	3.559	3	7.5	−2	1	1	−1.563	133.383	172.141	−6.845
V222	127.504	−5.182	3.093	5	7.25	−1	2	3	−1.731	101.888	231.806	−6.597
V226	103.042	−9.687	6.75	1.25	8.75	−2	4	1	−1.075	2761.001	715.442	−7.297
V231	113.094	−9.105	5.992	2.25	7.75	−2	3	1	−1.085	1992.813	528.81	−7.326
V243	125.557	−9.876	6.523	2.25	9	−2	5	1	−1.568	1356.918	372.469	−7.538
V245	116.326	−9.815	6.884	1.25	9.5	−2	5	1	−1.416	1933.915	514.511	−7.371
V247	132.565	−6.51	5.629	2.25	11	−1	6	1	−0.775	522.9	152.091	−7.308
V253	143.489	−7.077	4.417	2.25	10.95	−2	6	1	−1.538	184.545	53.315	−8.299
V254	129.75	−8.419	5.29	1.25	9.45	−2	5	1	−1.861	687.446	197.201	−7.324
V282	123.041	−6.679	3.535	3	10	−1	2	1	−0.581	408.384	110.962	−7.606
V291	134.153	−5.819	2.54	5	8	−2	2	2	−1.351	53.764	17.108	−7.099
V304	129.939	−5.218	2.33	4	9.5	−1	4	2	−0.865	146.26	23.945	−6.2
Standard range	7–200	−6.5–0.5	−2.0–6.5	0.0–6.0	2.0–20.0	−2–+2	1–8	1, 2, or 3 for low, medium, or high	−3.0–1.2	<25 poor, >500 great	<25 poor, >500 great	<−5

**Table 4 ijms-24-11390-t004:** The IC_50_ values of the top five compounds.

Compounds	Structure	IC_50_ Value (μM)
V111	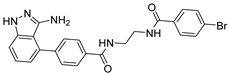	>20
V139	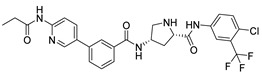	>20
V159	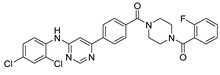	>20
V205	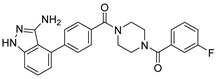	>20
V226	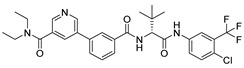	>20
V231	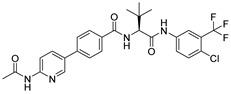	>20
V243	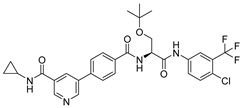	>20
V291	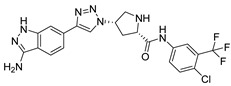	2.77 ± 0.56
V304	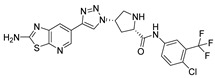	>20
Nilotinib	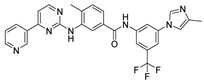	19.92 ± 1.27

## Data Availability

The data presented in this study are available in Appendix A.

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
