# Peer review of "Identification of a Putative SARS-CoV-2 Main Protease Inhibitor through In Silico Screening of Self-Designed Molecular Library"

_ijms, 2023, doi:10.3390/ijms241411390_

Round 1
Reviewer 1 Report
The paper by Liu et al contains a nice piece of work from the point of view of the design and in silico screening of potential inhibitors of the SARS-Cov-2 protease. The methods used are up-to-date and sound and the work is well done and well described. The use of ADMET prediction tools is something that should be highlighted since it is not yet common to see it in papers that seek to develop a possible drug candidate. I congratulate the authors for including it.
However, from the point of view of a Medicinal Chemist there is some information lacks that should be corrected to improve the quality of the manuscript:
1) The structures of Nilotinib and, if available, compound N3 should be included in the introduction for a better understanding of such a part.
2) More importantly, the authors select 9 compounds out of a virtual library of 320 compounds and biologically evaluate such 9 compounds without giving any details of their synthesis. Neither in the experimental part nor in the supplementary information there are such synthetic details. I consider it mandatory to include the synthesis description of such 9 compounds before point 2.6 including a synthetic scheme, reaction conditions, yields, and all the experimental details in the experimental part and spectra in the supplementary information. Otherwise, I cannot recommend the publication of this manuscript because no evidence shows that these 9 compounds were really synthesized.
Reviewer 2 Report
In the manuscript (ID: ijms-2441435) the authors identified novel potential inhibitor of Mpro by applying a virtual screening of hundreds Nilotinib structure-like compounds by using in silico and in vitro methods.
In the manuscript the authors should take under consideration the following points:
1) On the Figure 1 the caption on the figure should be more larger.
2) The authors write that they compiled database of more than 300 compounds, but they didn’t mention how the compounds were selected or build, no information about database or methods used to discover new compounds. The authors should mention that the structures of compounds are in supplementary materials.
3) The references to the software used should be included.
4) Do the authors check the ligand binding in the Mpro and the aminoacids that interact with ligand after docking?
5) The Figures 3-8 must be improved, especially part of the figure with showed interactions, captions are not readable.
6) On the Figure S1 are shown only 14 compounds and in the manuscript the authors wrote that 20 complexes was subjected to MD. It should be corrected.
7) The Figure 9 should be improved, the grey and black colors should be changed, especially Figure 9B is not readable and must be improved.
8) Why the authors show Figures of RMSD for 20ns? At the beginning of MD all system (protein-ligand complex) must reach the equilibrium and after that we can analyze the dynamics of the complex. RMSD is an indicator of the stability of the protein-complex and should be less than 2Å, but less than 3Å is still acceptable. The curve should show minimal fluctuations near the end of the simulation time. Otherwise, if the curve is still rising, that's usually an indicator that a longer simulation time is needed. So the authors need to check rmsd of all 100 ns of simulation to find if it is rising near the end of simulation.
9) What is IC50 of Nilotinib?
10) In the table 4 the inhibitory activity is shown only for 5 compounds. Why the authors didn’t show IC50 for the rest of 9 compounds selected for experimental study?
11) The methodology is not well described, especially molecular dynamics (what are parameters of MD) and induced fit docking (what are parameters). In the induced fit docking the authors didn’t describe if the aminoacids conformations are changed and conformation of ligand?
Reviewer 3 Report
The article "Identification of a Putative SARS-COV-2 Main Protease Inhibitor through In Silico Screening of Self-designed Molecular Library" is a nice article about the in-silico and in-vitro evaluation of a library of compounds against the COVID main protease Mpro. The article starts describing the improtance of this protease for the virus maturation and the main inhibitor found against it, then evaluate the interaction about the known inhibitor Nilotinib and the amino acids within the active site before analyzing the new 300 compounds via docking and molecular dynamics. Finally the article ends with the biological evaluation of best inhibitor candidates.
The in-silico part is well described, with a robust analysis and a nice presentation of data, even if a re-check of some words is needed. The biological evaluation is a bit vague, and should be described better as suggested in my following comments:
line 86: The app you used for protein preparation is Glid or Glide? Guess u missed the e.
line 92: Did you use any other software to evaluate the compounds Kd? The direct convertion of Glide binding energy score to Kd is quite inaccurate and usually other programs like Gromacs or Amber are used to evaluate Glide results and confirm the Kd. So can you please describe this convertion?
line 111: guess infused means induced.
line 188: Mpro is not superscript. The same in complexes in line 286. Can you please uniform the text for a more clear lecture?
line 297: ADME/T is a very important evaluation standard - in this sentence u mean "standard evaluation" maybe?
line 357: both the "gride" should be grid without e?
line 372: MMGBSA miss a -
In vitro assay (FRET) - I have a huge question about this method: since the assay is discontinous are you sure to be in "enzymatic linear phase"? Indeed, to be sure of your IC50 evaluation you should choose an incubation time for which in absence of inhibitor (positive control) the enzyme is still acting and didn't finish to catalyze all the substrate, nor is close to catalyze it all. It is important to be in linear phase of enzymatic activity for a nice evaluation of IC50. So I think you could add in the discussion or in methods a couple of sentences in which you describe the time needed by your enzyme to catalyze all the substrate at the described condition, and that you're always using the same enzyme concentration in every evaluation. I expect you realized this experiment in this way but couldn't find this explanation (that is essential) in the text. If you didn't use always the same enzyme concentration or didn't check the time for the full substrate hydrolysis I need to ask you this evaluation to be added to your manuscript.
bibliography: please check the bibliography since it is not uniform. For example citations 12-15 has the first letter of every word in Capital while many others not.
I recommend this article for publication on IJMS after minor revision.
Round 2
Reviewer 1 Report
Thank you for clarifying and addressing the issues that I found in your previous version. As I said, a nice paper that now can be published in its present form.
Reviewer 2 Report
The manuscript (ID: ijms-2441435) about identifying novel potential inhibitor of Mpro by using in silico and in vitro methods was improved according reviewer suggestions. However, the authors should improve the Figure 11 is unclear.
